# Identifying Algicides of *Enterobacter hormaechei* F2 for Control of the Harmful Alga *Microcystis aeruginosa*

**DOI:** 10.3390/ijerph19137556

**Published:** 2022-06-21

**Authors:** Bin Zhang, Ying Yang, Wenjia Xie, Wei He, Jia Xie, Wei Liu

**Affiliations:** 1School of Environmental Science and Engineering, Sun Yat-sen University, Guangzhou 510006, China; zhangb268@mail2.sysu.edu.cn (B.Z.); xiewj3@mail2.sysu.edu.cn (W.X.); hewei67@mail2.sysu.edu.cn (W.H.); xiej79@mail2.sysu.edu.cn (J.X.); 2School of Marine Sciences, Sun Yat-sen University, Zhuhai 519082, China; yangying6@mail.sysu.edu.cn; 3Southern Marine Science and Engineering Guangdong Laboratory (Zhuhai), Zhuhai 519080, China; 4Guangdong Provincial Key Laboratory of Environmental Pollution and Remediation Technology, Guangzhou 510006, China

**Keywords:** algicidal bacteria, prodigiosin, quorum sensing molecular, transcriptome

## Abstract

Eutrophication has become an increasingly serious environmental issue and has contributed towards an explosion in harmful algal blooms (HABs) affecting local development. HABs can cause serious threats to ecosystems and human health. A newly isolated algicidal strain, *Enterobacter hormaechei* F2, showed high algicidal activity against the typical HAB species *Microcystis aeruginosa*. Potential algicides were detected through liquid chromatograph–mass spectrometer analysis, revealing that prodigiosin is an algicide and PQS is a quorum sensing molecule. RNA-seq was used to understand the algicidal mechanisms and the related pathways. We concluded that the metabolism of prodigiosin and PQS are active at the transcriptional level. The findings indicate that *E. hormaechei* F2 can be used as a potential biological agent to control harmful algal blooms to prevent the deterioration of the ecological and economic value of water bodies.

## 1. Introduction

Harmful algal blooms (HABs) can cause serious damage to the culture and quality of aquatic environments [1]. The cyanobacterium *Microcystis aeruginosa* is a typical HAB species, which can produce toxic microcystins (MCs). Concentrations of MCs at 30 μg/L were found to cause damage to mice and zebrafish [2,3]; however, the concentrations of these compounds in numerous surface water tests have exceeded this level [4,5,6].

HABs can be treated by physical, chemical, and biological methods, among which biological treatment utilizing microbial agents is effective at inhibiting their growth and reducing eutrophic substances in natural water bodies. Algicidal bacteria act on algal cells via direct contact and indirect attack. The former involves directly attacking algal cells through the direct contact of algal surfaces, whereas the latter involves killing algal cells by secreting algicides [7,8]. Due to the high bacterial diversity and difficulty in separating algicidal substances, many algicides are still unknown, and only a few algicides have been identified, which include peptides [9,10], alkaloids [11,12], and lipids [13,14]. One common algicide is prodigiosin, a red pigment belonging to the antibiotic family, and is an efficient algicide [15,16] against *Phaeocystis globosa* [17], red tide phytoplankton [18], and *M. aeruginosa* [19,20,21]. Several extracellular enzymes have also been identified as potent algicides. The activity of extracellular enzymes of white rot fungi *Phanerochaete chrysosporium* varied with the algicidal process, and different enzymes functioned in different algicidal phases, for example, laccase and MnP mainly function in the early and late stages, respectively [22]. 

Algicidal bacteria degrade algal cells by destroying their cell morphology through cell membrane and cell wall damage, as well as affecting their physiological activities [23]. Some studies have revealed that more algal cells are propidium iodide-stained after co-culture with algicides, indicating the effect of algicides on cell membrane permeability, thereby allowing the entry of foreign substances into the cells, and consequently damaging cellular organelles [24]. Algicidal bacteria can also reduce the photosynthesis efficiency and antioxidant activity of algal cells by destroying the photosynthesis system that absorbs light and suppressing electron transfer in photosynthesis reactions [17,19,25]. Furthermore, algicidal bacteria can increase superfluous reactive oxygen species content in algal cells and ultimately cause algal cell death [19,26].

Previous studies on algicidal bacteria have mainly focused on purifying and identifying algicides or methods to promote algicidal efficiency. Studies on algicidal mechanisms have primarily focused on algal cell damage, whereas studies based on molecular biology perspectives are limited. Kwon et al. [27] found a gene cluster related to prodigiosin secretion with an algicidal effect in *Hahella chejuensis* and *Streptomyces coeclicolor A3*. By using transcriptional analysis, Zhang et al. [28] indicated that the algicidal activity of *Brevibacillus laterosporus* was performed by suppressed photosynthesis and oxidative phosphorylation of *M. aeruginosa*, which could block electron transport and affect energy absorption of the cyanobacterium.

The aim of this study was to identify the algicides secreted by *Enterobacter hormaechei* F2, an algicidal bacterium isolated in a previous study, and determine the relevant mechanisms of bacterial gene expression during the algicidal process using transcriptomic analysis. Finally, transmission electron microscopy (TEM) and scanning electron microscopy (SEM) were used to observe morphological changes in algal cells during the algicidal process. The results could provide useful information for the bio-control of HABs, which pose a severe environmental threat.

## 2. Materials and Methods

### 2.1. Growth Conditions of Algicidal Bacteria and Cyanobacterial Cells

The algicidal bacteria *E. hormaechei* F2 was isolated from the Pearl River, Guangzhou, China. The strain was cultured in a Luria–Bertani medium (LB medium: 5 g yeast extract, 10 g tryptone in L distilled water, pH 7.0) in an incubator shaker at 150 rpm and 37 °C for 48 h. The experimental cyanobacterium, *M. aeruginosa* (FACHB-315), was provided by the Freshwater Algae Culture collection at the Institute of Hydrobiology of Chinese Academy of Sciences (FACHB, Wuhan, China), and was cultured in BG11 medium (BG11 medium: 1.500 g NaNO_3_, 0.040 g K_2_HPO_4_, 0.001 g EDTA·2Na, 0.075 g MgSO_4_·7H_2_O, 0.036 g CaCl_2_·2H_2_O, 0.006 g ferric ammonium citrate, 0.006 g citric acid, 0.020 g Na_2_CO_3_, 1 mL A_5_+Co solution in 1 L distilled water, pH 7.1). The A_5_+Co solution was prepared by dissolving the following in 1 L of distilled water: 2.860 g H_3_BO_3_, 1.810 g MnCl_2_·4H_2_O, 0.049 g Co(NO_3_)_2_·6H_2_O, 0.222 g ZnSO_4_·7H_2_O, 0.079 g Na_2_MoO4·2H_2_O, 0.390 g CuSO_4_·5H_2_O. The cyanobacterial culture was kept at 25 °C under a 15:9 h light-dark cycle, with a light intensity of 2500 Lux.

### 2.2. Algicidal Activity

To determine the algicidal activity of *E. hormaechei* F2 on *M. aeruginosa*, *E. hormaechei* F2 was inoculated in 100 mL LB medium and grew to a stationary phase for 48 h in a shaker at 150 rpm. Then, the bacterial culture was added to the cyanobacterial cultures at a concentration of 2.0% (final concentration at 2 × 10^5^ CFU/mL), whereas the cyanobacterial cultures with the same concentration of LB medium were established as the control group. The cell density of *M. aeruginosa* was determined by measuring the chlorophyll *a* (Chl *a*) concentration using the acetone method [29]. 

*Microcystis aeruginosa* cells were counted by a hemocytometer plate according to Equation (1).
(1)$$\mathrm{Cells}/\mathrm{mL}=\frac{N}{100}\times 400\times 10^{4}\times \mathrm{Dilution}\;\mathrm{factor}$$
where *N* represents the cell numbers in 100 small squares.

The algicidal ratio was calculated according to Equation (2).
(2)$$\mathrm{Algicidal}\;\mathrm{ratio}=\frac{Nc-Ne}{Nc}\times 100$$
where *Nc* and *Ne* represent the algal chlorophyll *a* concentration in the control and experimental groups, respectively. The algicidal ratio was determined after inoculation for 4 days.

Prodigiosin (Tokyo Chemical Industry, Tokyo, Japan), PQS (Bidepharm, Shanghai, China) and phenazine (Bidepharm, Shanghai, China) were dissolved in dimethyl sulfoxide for algicidal activity. The mother liquor concentration was 2 mol/L, with filter sterilization. Equal amounts of dimethyl sulfoxide were added to the control group for these experiments.

Laccase activity was performed by the ABTS method [22].

### 2.3. Nucleic Acid-Related Manipulation

Genomic DNA was extracted with a DNA Kit (Omega, Hampton, NH, USA, D3195-01). PCR amplification was performed using Phusion High-Fidelity DNA Polymerase (Thermo Scientific, Waltham, MA, USA, lot:00528748). DNA fragment elution was performed using a Gel Extraction Kit (Omega, D2500-02) and/or a Cycle Pure Kit (Omega, D6492-02). For the Southern blot assay, restriction enzymes used for the digestion of genomic DNA were from NEB (New England Biolabs (Beijing) Ltd., Beijing, China). The Labeling and Detection Starter Kit I (Roche, Basel, Switzerland, 11745832910) was used to label PCR amplified fragments as a probe. Amersham Hybond TM-N+ (GE Healthcare, Chicago, IL, USA, RFN303B) membranes were used for blotting. NBT/BCIP Stock Solution (Roche, 11681451001) was used for probed band detection. 

For the transcriptome assay, the co-cultivation of *E. hormaechei* F2 and *M. aeruginosa* on 0 d were used as controls, and 4 d and 7 d as the experimental groups. The cells were collected by centrifugation (5000× *g*, 10 min, 4 °C) and total RNA extraction was performed by a Qiagen RNeasy Mini kit (74104). The Ambion^®^ TURBO DNA-free™ kit (Invitrogen, Waltham, MA, USA, AM1907) was used to remove the contaminating DNA from the RNA preparations. The TransScript^®^ First-Strand cDNA Synthesis Super Mix (Transgen, Beijing, China, AT301-02) was used for cDNA synthesis. The differentially expressed genes (DEGs) screening conditions were as follows: log2|fold change| > 1, *p*-value < 0.05.

The identified genes were annotated using the Kyoto Encyclopedia of Genes and Genomes (KEGG) Compound database (http://www.kegg.jp/kegg/compound/ (accessed on 10 February 2020)) and mapped to the KEGG Pathway database (http://www.kegg.jp/kegg/pathway.html (accessed on 10 February 2020)), as well as the Gene Ontology (GO) database (http://geneontology.org/ (accessed on 10 February 2020)). Pathways with significantly regulated genes and metabolites mapped were then fed into metabolite set enrichment analysis (MSEA), and their significance was determined using hypergeometric test *p*-values.

For real-time qPCR, the sample collection and RNA extraction were the same as the transcriptome assay. We used PowerUp™ SYBR^®^ Green Master Mix (Applied Biosystems, Waltham, MA, USA, A25742), and the reaction (95 °C 5 min, (95 °C 10 s, 60 °C 30 s) 30 cycles, 95 °C 15 s, 60 °C 60 s, 95 °C 15 s) ran on the QuantStudio 6 Flex Real-Time PCR System (Thermo Fisher Scientific, Waltham, MA, USA). Next, 2^−ΔΔCt^ was used to calculate the relative fold change. The primers used for qRT-PCR analysis are listed in Appendix A.

### 2.4. Extraction and Purification of the Algicidal Compound

The algicidal bacteria was grown in a 1 L flask with LB medium in a shaker (150 rpm, 37 °C) and then centrifuged (5000× *g*, 10 min, 4 °C) to obtain the cell-free medium. Then, the solutions were concentrated to 100 mL using a rotary evaporator (40 °C), supplemented with 65 mL of 95% ethanol, and placed into the fridge (4 °C) overnight. Then, the culture was centrifuged (5000× *g*, 10 min, 4 °C) to obtain the supernatant. Petroleum ether, ethyl acetate, n-butyl alcohol, and chloroform were utilized as extractants and were added with the extract to the cyanobacterial culture at a concentration of 1 g/L to test the algicidal effect. The n-butyl alcohol extract of the cell-free medium showed the strongest algicidal activity. Then, the n-butyl alcohol extract was separated with a silica gel chromatographic column via a degraded chloroform and methanol series (*v/v* = 1:0, 5:1, 2:1, 1.5:1, 1:1, 1:1.5 1:2, 1:5, 0:1) at a rate of 0.75 mL/min, and each fraction was evaporated and added to the cyanobacterial culture at a concentration of 1 g/L to test the algicidal activity. The fractions were defined as follows: extract 1 (*v/v* = 1:0), extract 2 (*v/v* = 5:1), extract 3 (*v/v* = 2:1), extract 4 (*v/v* = 1.5:1), extract 5 (*v/v* = 1:1), extract 6 (*v/v* = 1:1.5), extract 7 (*v/v* = 1:2), extract 8 (*v/v* = 1:5), extract 9 (*v/v* = 0:1). Among these fractions, extract 2 (*v/v* = 5:1) and extract 7 (*v/v* = 1:2) exhibited strong algicidal effects.

### 2.5. Liquid Chromatograph–Mass Spectrometer (LC–MS) Analysis

Extract 2 and 7, prodigiosin (Tokyo Chemical Industry, Tokyo, Japan) and PQS (Bidepharm, Shanghai, China) were dissolved in methanol, and the solutions were filtered with a 0.22 μm filter membrane. Compound separation was performed with a reverse-phase chromatographic column (Agilent XB-C18, 4.6 mm × 50 mm, 2.6 μm) at 40 °C, with mobile phase A consisting of 0.1% HCOOH, mobile phase B comprising 0,1% HCOOH-CAN, and a flow rate of 1.2 mL/min. The effluent was transferred to the mass spectrometer with an electro-spray ionization source. The mass range was between 105 and 1500 m/z. The calculated method referred to Liu et al. [30].

### 2.6. Transmission Electron Microscopy

Cyanobacterial cells were incubated with the bacterial culture for 0, 4, and 7 d, and then prepared for TEM. Next, 30 mL of cyanobacterial culture was collected (5000× *g*, 10 min, 4 °C) and fixed with 2.5% glutaraldehyde (*v/v*) for 24 h. The sample was then rinsed three times with PBS buffer and fixed with 1% OsO_4_ for 2 h. The samples were then rinsed with the PBS buffer three times, dehydrated in a degraded ethanol series (30%, 50%, 70%, 80%, 90%, 95% and 100%), and then stored in tertiary butanol for 20 min. The samples were embedded with epoxy resin and acetone (*v*:*v* = 3:1) for 3 h and then embedded for one night at 70 °C. The sections (70–90 nm) were obtained with an ultramicrotome (LEICA-EM-UC7) and stained with lead citrate and uranium-citric for 5 min. The samples were viewed by TEM (H7650).

### 2.7. Scanning Electron Microscopy

Cyanobacterial cells were incubated with bacterial culture for 0, 4, and 7 d and then prepared for TEM. Here, 30 mL of cyanobacterial culture was collected (5000× *g*, 10 min, 4 °C) and fixed with 2.5% glutaraldehyde (*v/v*) for 24 h. The sample was then rinsed with PBS buffer and fixed with 1% OsO_4_ for 2 h. The samples were rinsed with PBS buffer three times, dehydrated in a degraded ethanol series (30%, 50%,70%, 80%, 90%, 95% and 100%), and then stored in tertiary butanol for 20 min. Next, the samples were treated with ethanol and isoamyl acetate solution (*v/v* = 1/1) for 30 min and stored in isoamyl acetate overnight. The samples were critical-point-dried and coated with gold-palladium and finally imaged by SEM (SU8010).

### 2.8. Statistical Analysis

In the transcriptome analysis, unsupervised principal component analysis (PCA) was performed using the statistical function prcomp in R (www.r-project.org (accessed on 3 March 2020)). The data were unit variance-scaled before unsupervised PCA. The hierarchical cluster analysis (HCA) results of the samples and metabolites were presented as heatmaps with dendrograms, whereas Pearson correlation coefficients (PCC) between the samples were calculated using the cor function in R and presented as heatmaps. Both HCA and PCC were performed using the R package ComplexHeatmap. For HCA, the normalized signal intensities of the genes and metabolites (unit variance scaling) were visualized as a color spectrum.

In the algicidal activity analysis, the figures and statistics are based on mean value ± standard error (S.E.). Significance analysis of difference was performed by paired Student’s *t*-test.

## 3. Results and Discussion

### 3.1. Algicidal Effects of the Strain

An algicidal bacterium, which was identified and named as *E. hormaechei* F2 in a previous study, showed effects on the damage or death of cyanobacterium cells (Figure 1A). The cyanobacterial solution appeared green at the beginning of bacteria-cyanobacterium co-cultivation, and the color of the cyanobacterial solution faded slightly at 4 d and became nearly colorless at 7 d of co-cultivation. The color change during the co-cultivation period suggested the algicidal effect of *E. hormaechei* F2 on *M. aeruginosa*.

Electron microscopy was used to observe the morphological changes in the cyanobacterium under the influence of algicidal *E. hormaechei* F2 (Figure 1B). Before co-cultivation (0 d of co-cultivation), cyanobacterial cells exhibited a spherical shape with a smooth surface; however, after 4 d of co-cultivation, cyanobacterial cells shrunk, and the SEM images showed the adherence of bacterial cells to the cyanobacterial cell surface. After 7 d of co-cultivation, the cyanobacterial surface became significantly rugose and the bacteria adhered to the cyanobacterial cells, which demonstrated direct contact algicidal activity [31,32].

Changes inside of the cyanobacterial cells are shown by the TEM images (Figure 1C). Before co-cultivation, the cells had a complete membrane structure, the cytoplasm was evenly distributed, and the thylakoid structure was closely arranged. With an increase in time, the structure of the cell membrane became irregular, vacuoles appeared in the cells, and thylakoids were loosely and irregularly arranged. By the end of co-cultivation, the cyanobacterial cells became smaller and the cytoplasm and thylakoids were reduced, indicating the death of the cyanobacterial cells.

### 3.2. Detection of Algicides

#### 3.2.1. Separation and Identification of Algicides

Algicidal bacteria can secrete various algicidal compounds [7,8,19,20,21]. To investigate the algicides secreted by *E. hormaechei* F2, components from the cultivation medium of the strain were extracted using nine different volume ratios of chloroform and methanol, and the algicidal efficiency of each extract were further tested. As shown in Figure 2A, Extracts 2, 5, 6, and 7 demonstrated higher inhibition efficiencies than other extracts, which were 59.54%, 54.60%, 57.49%, and 59.54%, respectively. However, the inhibition efficiencies of all the extracts were lower than those of the co-cultivation system, suggesting that the effective algicidal performance of *E. hormaechei* F2 (84.2%) was a combined result of different algicides and mechanisms. 

To further explore the potential algicides in extracellular substances, liquid chromatograph–mass spectrometer analysis (LC–MS) was employed to detect the chromatograms of Extracts 2, 5, 6, and 7, using the widely reported algicide prodigiosin and AQs family (2-alkyl-4-quinolone family) [17,25,27,33] as standard substances. A characteristic peak of PQS-like (2-heptyl-3-hydroxy-quinolone, *Pseudomonas* quinolone signal) substances, which belong to the AQs family, was found in Extract 2, indicating the existence of PQS. The presence of prodigiosin was also shown in the chromatogram of Extract 7 (Figure 2C). Neither a PQS-like substance nor prodigiosin was found in Extract 5 and 6, suggesting the presence of an unknown algicide. 

#### 3.2.2. Algicidal Effects of Identified Substances

##### Prodigiosin as a Potential Algicide

The chromatogram of the extracellular substances demonstrated that PQS and prodigiosin are potential algicidal compounds of *E. hormaechei* F2, and they were constitutively expressed and produced. Further studies were conducted to determine the algicidal effects of these compounds. However, unlike other reported AQs families, such as HHQ (2-heptyl-4-quinolone) [34,35], the growth of *M. aeruginosa* was not affected after a 4 d treatment with PQS (Figure 3A). Prodigiosin, the algicide identified in Extract 6, belonged to a family of red pigments [17,25,27,33]. It has also been reported that phenazine-like components can control red tide organisms [36]. Prodigiosin showed algicidal activity at a concentration of 0.5 μM (Figure 3), as indicated by the fading of the cyanobacterial color and the significant decrease in cyanobacterial cell concentrations from 1 × 10^7^ to 5 × 10^5^ cells/mL. 

SEM images revealed that with prodigiosin treatment, the cyanobacterial cells were severely deformed when compared with the control cells (Figure 3B). TEM images showed the inner damage of the cyanobacterium; when the cyanobacterial cells were treated with prodigiosin, the cell wall remained intact, the peptidoglycan layer remained intact (the cell wall was maintained as a double-layer structure), chromatin contracted, and the carboxysome structure was maintained (Figure 3C). These phenomena are similar to the previously observed algicidal effects of the strain, which suggests that prodigiosin is a potential algicide of *E. hormaechei* F2.

##### PQS Might Act as a QS Molecule That Regulates Algicidal Genes and Enzymes

The direct algicidal effect of PQS from *E. hormaechei* F2 was not observed in this study; however, a previous study proposed that PQS might act as QS signaling molecule that can regulate strain algicidal activity [35]. Therefore, we tested this hypothesis by adding pure PQS into the bacteria-cyanobacterium co-cultivated system. The results of the cyanobacterial cell concentration and chlorophyll *a* content analyses both showed that the addition of the PQS-like substance could effectively enhance the algicidal effect of *E. hormaechei* F2 (Figure 4A,B). The fading of the green color was accelerated by an increase in the dosage of the PQS-like substance. As shown in Figure 4C, treatment with 2 μM of the PQS-like substance from *E. hormaechei* F2 for 4 d resulted in the cyanobacterial solution becoming almost transparent, compared with the control. Furthermore, there was a significant reduction in both the cyanobacterial cell concentration and chlorophyll *a* content, suggesting a better algicidal effect than in other groups at 4 d, and a more significant effect at 7 d. In the subsequent experiment, 2 μM PQS was used to study the effects of this substance and mechanism.

As QS molecules generally promote bacterial growth [37], the effect of PQS on *E. hormaechei* F2 growth was assessed. Notably, the growth of *E. hormaechei* F2 was slightly suppressed after 16 h of incubation with PQS treatment (Figure 5A), implying that PQS promotes cyanobacterial inhibition by regulating gene expression rather than by increasing bacteria biomass. To verify this assumption, enzymatic activity and qRT-PCR analyses were performed. Laccase, which was reported to be an important algicidal enzyme [22], showed significantly higher activity when co-incubated with *M. aeruginosa*, suggesting the involvement of laccase in the algicidal processes of *E. hormaechei* F2. With the presence of PQS at a concentration of 2 μM, the activity of laccase became significantly higher, indicating that PQS can increase the activity of the enzyme, which promoted the algicidal activity of *E. hormaechei* F2 (Figure 5B).

Three algicide-related genes, including K00059, K00652, and K01657, were chosen for an expression test [17,18,19,27]. The results showed that in the absence of PQS, the expression level of K00059 did not change from 4 to 7 d. However, K00652 and K01657 were up-regulated at 7 d relative to levels at 4 d. When PQS was added, the expression level of these genes remained similar to that in the absence of PQS on day 4 but were highly up-regulated at 7 d, indicating that the regulatory effect of PQS was most pronounced after 4 d. In summary, PQS participates in the algicidal activity as a QS molecule that regulates algicide-related enzyme activity and gene expression.

### 3.3. Transcriptome Analysis Reveals the Algicidal Mechanism of the Strain

#### 3.3.1. Enrichment of Differentially Expressed Genes (DEGs) during the Algicidal Process

To better understand the algicidal mechanism and the algicide-related pathways, a transcriptome analysis of gene expression in the algicidal bacterium *E. hormaechei* F2 in the co-cultivation system was performed. DEGs were assessed among 0 d, 4 d, and 7 d timepoints. 

The overall distribution trend among all the samples can be observed using PCA, and the possible discrete points can be determined. PCA score plots are presented in Appendix A, the first principal component accounted for 61.49% and the second principal component for 12.14% of the variance; these two principal components contributed over 70%, which represent the characteristics of the data. The PCA score plots showed samples at 4 d (blue dots) and were distinguished from samples at 0 d (red dots) and 7 d (yellow dots), which indicated a specificity of samples at 4 d (Appendix A). Between 0 d and 4 d, 1023 genes were differentially expressed, with 614 up-regulated and 409 down-regulated. In contrast, 74 up-regulated and 63 down-regulated genes were detected between 4 d and 7 d, indicating that *E. hormaechei F2* was more active in the early stage, based on a higher number of DEGs identified between 0 d and 4 d (Appendix A). 

Hierarchical clustering analysis is a common method for data mining and statistics. As shown in the heatmaps and dendrogram (Appendix A), genes or samples with similar variation trends in the absolute values of expression fold change extent were positioned closer together. As for the samples, 4 d and 7 d samples were clustered into one branch, indicating that the variation trends of DEGs were similar.

To further investigate the algicidal functions of *E. hormaechei* F2, a gene set enrichment analysis (GSEA) was performed based on the GO database. The GO terms of the identified DEGs at 0 and 4 d were enriched in the nitrate reductase complex of the cellular component category, and ATP synthesis coupled proton transport and phenylacetate catabolic process were enriched in the biological process category (Figure 6A). However, the GO terms of the identified DEGs between 0 and 7 d were enriched in catalytic activity, nitrate reductase activity, and oxidoreductase activity in the molecular function category; nitrate reductase complex in the cellular component category; and phenylacetate catabolic process, oxidation-reduction process, and nitrate metabolic process in the biological process category (Figure 2B). Furthermore, the GO terms of the identified DEGs at 4 and 7 d were enriched in proton-transporting ATP synthase activity (rotational mechanism) in the molecular function category; proton-transporting ATP synthase complex, catalytic core F (1) in the cellular component category; and ATP synthesis coupled proton transport and peptide transport in the biological process category (Figure 2C). In summary, the results of the GO analysis suggest that energy metabolism and aromatic compound metabolism are involved in the algicidal process. Similarly, the KEGG analysis showed that signaling system, energy metabolism, and aromatic amino acid metabolism pathways were crucial to the algicidal activity (Appendix A).

#### 3.3.2. Comparative Genome and RNA Seq Analysis to Decipher the Associated Pathway

The enrichment analysis demonstrated that the signaling system, energy metabolism, and aromatic amino acid metabolism participated in the algicidal process. The identified algicide-related substance (i.e., PQS and prodigiosin) were the final products of the pathway of prodigiosin biosynthesis (map00333) and phenazine biosynthesis (map00405) pathways. To reveal the changes in these pathways at the transcriptional level, transcript profiling of DEGs was conducted (Figure 7). Similar to the analysis of DEGs, most were upregulated at 4 d of co-cultivation and slightly down-regulated at 7 d of co-cultivation, further suggesting that the fourth day is the most active in the algicidal process in terms of the transcriptional level.

Energy metabolism-related genes were upstream of the pathway, indicating an increasing extracellular energy metabolism, such as that related to sucrose, cellobiose, and D-glucose, which were most likely from the breakdown of cyanobacterium. With the enhanced downstream transcriptional activity, phenylalanine, tyrosine, and tryptophan metabolism showed upregulation, which suggested the potential algicide-related compounds in the pathways.

Comparative genomic analysis between *E. hormaechei* F2 and other algicidal bacteria (*H. chejuensis* KCTC2396, *Zobellia galactanivorans* DsijT, *Mangrovimonas yunxiaonensis* LY01, and Rhodobacteraceae bacterium PD-2) was employed to simplify the data (Appendix A). The results showed that the ortholog genes K00652 and K00059, reported as algicide-related genes that are related to the biosynthesis of prodigiosin and belong to the family of red pigments, were produced as algicides by bacteria [17,18,19,27]. Furthermore, phenazine-like pigments could control red tide organisms [36], which were observed in the transcript profiling (Figure 7), indicating that this could be an algicide of *E.*
*hormaechei* F2. 

Interestingly, phenazine-like pigments have been reported to control red tide organisms [36], and this is a product of the same pathway as PQS (Figure 6). The algicidal effect of phenazine was also assessed, which showed marked algicidal effects, as the color of the cyanobacterium solution almost completely faded after treatment (Appendix A). As indicated by red arrows in Appendix A, the cyanobacterial cell wall ruptured and the cytoplasm leaked, which are typical features of cell necrosis [10]. This result suggested that phenazine might be an important algicide for the strain.

The algicidal process of *E. hormaechei* F2 can be summarized as follows: (1) prodigiosin and phenazine may act as algicides of *E. hormaechei* F2. (2) PQS, which might act as a QS molecule, improved algicidal activity by enhancing gene expression and enzyme activity of *E. hormaechei* F2. (3) Simultaneously, *E. hormaechei* F2 adhered to cyanobacterial cells, resulting in the disruption of the cyanobacterial cell membrane in the next 4 d, thereby causing a loss of selective permeability. (4) Chlorophyll *a* content decreased, which could affect the photosynthetic efficiency. (5) Finally, the algal cells were fragmented into small molecules that can be used as nutrient sources.

## 4. Conclusions

In this study, *E. hormaechei* F2 showed significant algicidal activity against *M. aeruginosa* by secreting algicides. Subsequent analyses demonstrated that the potential algicides were PQS and prodigiosin, which could be detected in the extracellular substances of the strain. Prodigiosin showed high algicidal activity, and this phenomenon was also similar to the algicidal effect of the strain. Meanwhile, PQS did not show direct algicidal activity but could act as a QS molecule that regulates enzyme activity and gene expression. The transcriptome data suggested that throughout the entire algicidal process, amino acids and energy metabolism, a two-component system, the biosynthesis of secondary metabolites and quorum sensing participate in algicidal activity. PQS and prodigiosin are, thus, algicide-related compounds of the strain. The identification of algicides from *E. hormaechei* F2 and an understanding of the potential mechanism underlying their effects could promote our understanding of the performance of this algicidal bacteria, providing an effective way to control HABs, which pose a severe threat to inland aquatic ecosystems and human health.

## Figures and Tables

**Figure 1 ijerph-19-07556-f001:**
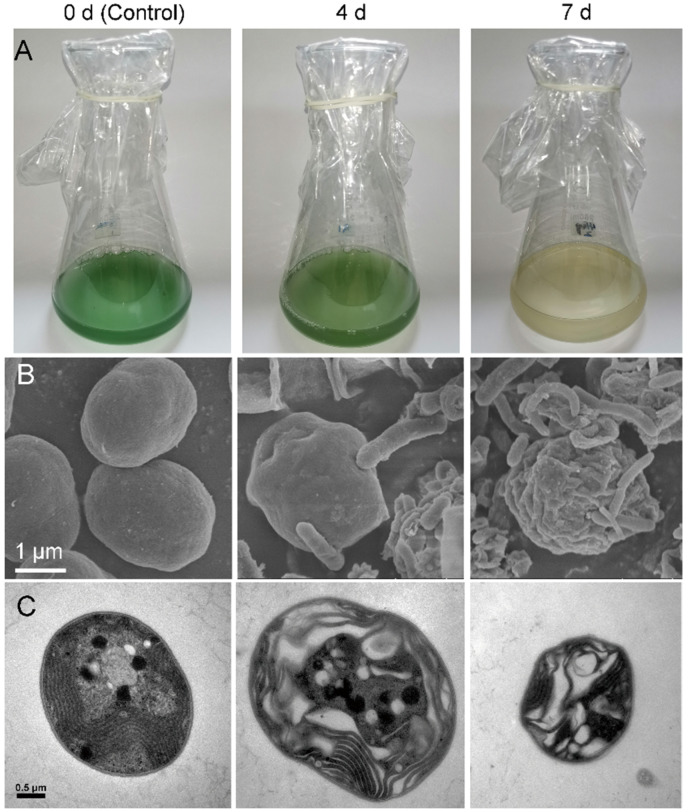
Electron microscopic images of *Microcystis*
*aeruginosa* co-incubated with *Enterobacter hormaechei* F2. (**A**) Phenotype-based images of algicidal effects. (**B**) Scanning electron microscopy images, bar = 1 μm; to enable easier distinguishing of bacteria from algae, the photo was processed in color, with cyanobacterial cells in green and bacteria in yellow. (**C**) Transmission electron microscopy images, bar = 0.5 μm.

**Figure 2 ijerph-19-07556-f002:**
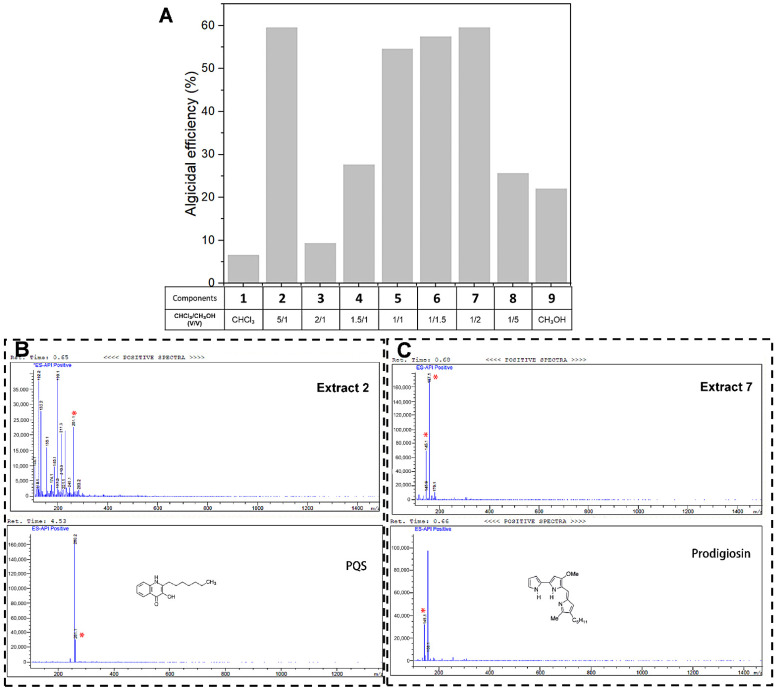
Algicidal effect of extracellular substances and potential algicides. (**A**) Algicidal efficiency of *Enterobacter hormaechei* F2 extracellular substances from different extracts. (**B**) LC–MS analysis to detect the presence of 2-heptyl-3-hydroxy-quinolone (PQS). (**C**) LC–MS analysis to detect the presence of prodigiosin; “*” indicates characteristic peaks of similar retention times.

**Figure 3 ijerph-19-07556-f003:**
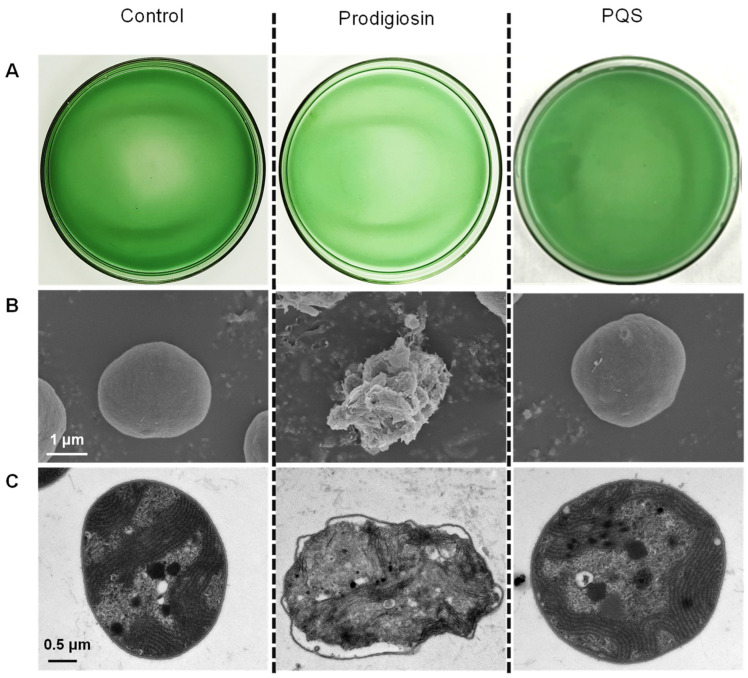
Algicidal activity of prodigiosin and 2-heptyl-3-hydroxy-quinolone (PQS). (**A**) Phenotype images that were taken from the bottom of the flask; (**B**) scanning electron microscopy images, bar = 1 μm; (**C**) transmission electron microscopy images, bar = 0.5 μm.

**Figure 4 ijerph-19-07556-f004:**
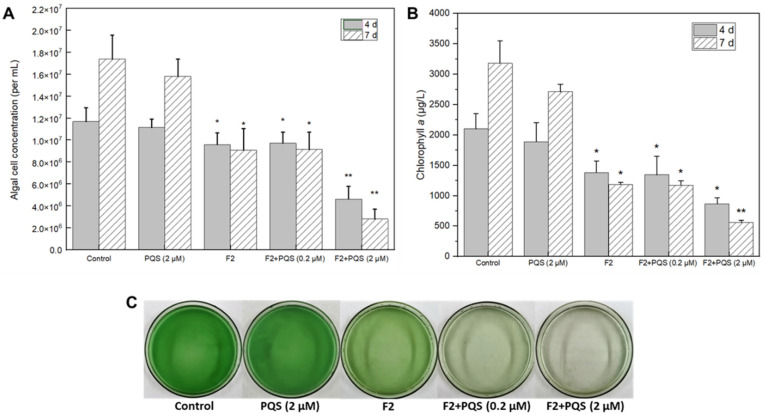
Algicidal activity of PQS from *Enterobacter hormaechei* F2. (**A**) Cyanobacterial cell concentration; (**B**) chlorophyll *a* content; (**C**) phenotype images that were taken from the bottom of the flask. “*” indicates a significant difference (0.01 ≤ *p* value ≤ 0.05), “**” indicates a very significant difference (*p* value ≤ 0.01).

**Figure 5 ijerph-19-07556-f005:**
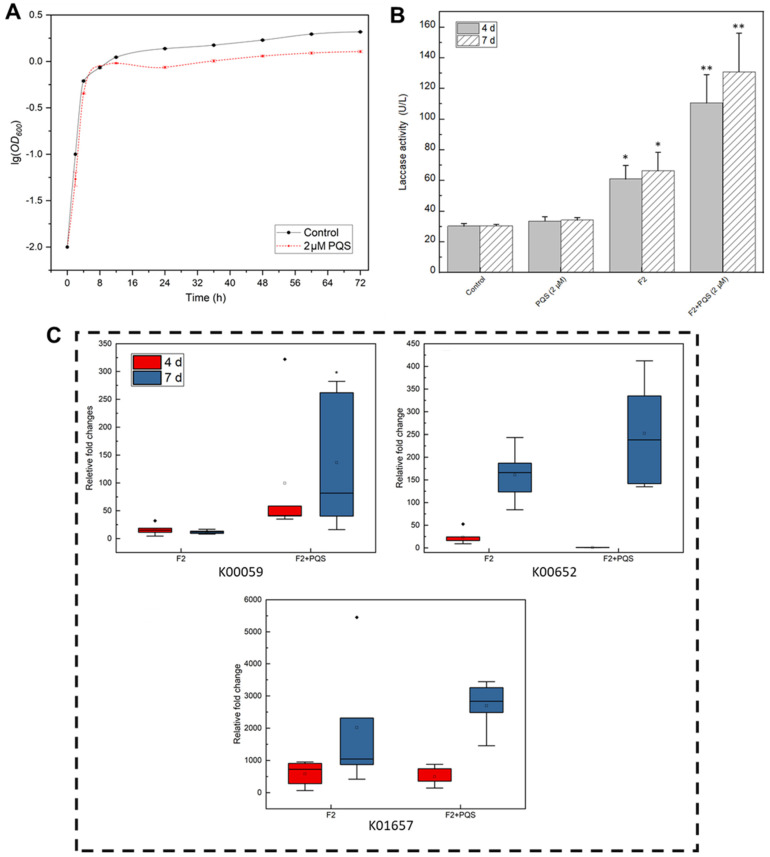
Effects of PQS from *Enterobacter hormaechei* F2. (**A**) Growth curve of *E. hormaechei* F2; the curve was prepared by plotting the logarithmic values of *OD*_600_ (optical density at 600 nm) vs. incubation time. Mean ± S.E. (**B**) Laccase activity in *E. hormaechei* F2; “*” indicates a significant difference (0.01 ≤ *p* value ≤ 0.05), “**” indicates a very significant difference (*p* value ≤ 0.01). (**C**) Relative gene expression fold-change, as calculated based on 16S rRNA as a housekeeping control.

**Figure 6 ijerph-19-07556-f006:**
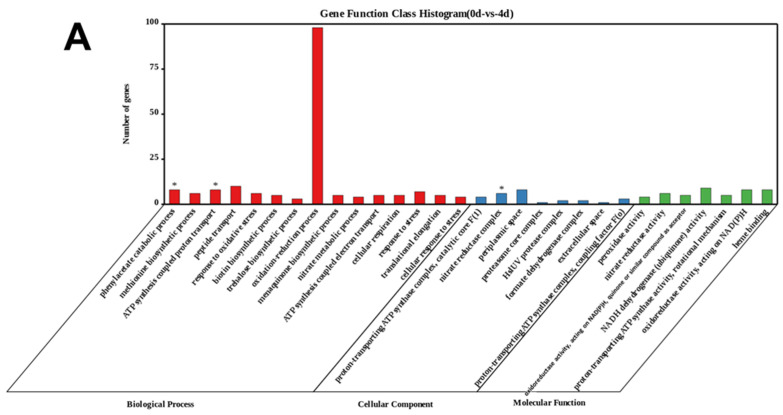
Gene Ontology (GO) functional analysis of differentially expressed genes (DEGs) for (**A**) 0 vs. 4 d (**B**) 0 vs. 7 d, and (**C**) 4 vs. 7 d. “*” indicates a significant enrichment of the GO term or KEGG pathway (*p*-value ≤ 0.05).

**Figure 7 ijerph-19-07556-f007:**
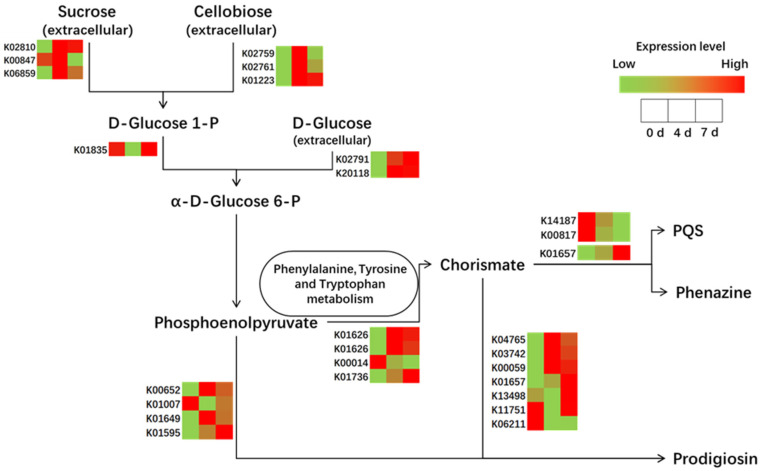
Transcript profiling of differentially expressed genes (DEGs). The color of the grids within a row indicates expression value of a DEG at 0, 4, and 7 d.

## Data Availability

The genome sequence of the bacteria was uploaded to the NCBI database (http://www.ncbi.nlm.nih.gov/nuccore/CP047570.1 (accessed on 15 October 2021), GenBank accession number CP047570.1).

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
