# Peer review of "Identifying Algicides of Enterobacter hormaechei F2 for Control of the Harmful Alga Microcystis aeruginosa"

_ijerph, 2022, doi:10.3390/ijerph19137556_

Round 1
Reviewer 1 Report
The manuscript entitled “Identifying algicides of Enterobacter hormaechei F2 for control of the harmful alga Microcystis aeruginosa” studies how the bacterial strain F2 and the algicidal substances it produces damage M. aeruginosa FACHB-315 cells. The topic is relevant and responds to the need to understand the mechanisms underlying the algicidal activity of some of the best-known compounds such as Prodigiosin and PQS. This study is particularly interesting because 1) there is few information about the effect of these compounds on M. aeruginosa, the most common bloom-forming cyanobacteria in freshwater, and 2) this is one of the first studies to perform transcriptomic analysis to unmask the mechanisms that explain the algicidal activity of certain bacteria on cyanobacteria.
However, even considering certain novelty, the interest of the results and the merit of the microscopy images, the paper has too many weaknesses. Therefore, in my opinion this article should be rejected for publication in the International Journal of Environmental Research and Public Health. Below I highlight key shortcomings, questions and remarks.
Concern#1: The English is poor, sometimes difficult to read, and there are quite a few grammatical errors, so I recommend proof-reading by native speakers.
Concern#2: One of the most important weaknesses of the manuscript is the poorly elaborated and explained methodology. Many crucial data are missing, making it difficult to understand how and why the experiments are performed. The most striking case is that they provide almost no information on gene expression and transcriptomic analysis. For example, compare with the following paper on the topic recently published in the same journal: Zhang et al., 2021. Other important missing information is how the authors performed the laccase activity assays or how they calculated the number of cells (Figure 4). Nothing from line 256-168 about adding PQS to the co-culture is explained in the material and method. New hypotheses appear throughout the article, but these should be highlighted and based on the main objectives at the end of the introduction.
Concern#3: The study is missing many important references, including the previous study in which the authors isolated Enterobacter hormaechei strain F2. There are many other references that are key to explaining the results but do not appear. For example, the earlier study where a Prodigiosin-producing bacterium was found to lyse M. aeruginosa cells (Yang et al., 2013), or the above commented study focused on transcriptomic analyses (Zhang et al. 2021). Also, the authors provided references focused on marine environments (i.e. L48-55), while the target cyanobacterium is mainly problematic and common in freshwater. Microcystis is the most studied bloom-forming genus of cyanobacteria, and there are more than 50 studies demonstrating the algicidal activity of various bacteria and compounds on it (for instance, see the last review on the topic: Yang et al., 2020). In my opinion, the authors should focus more on freshwater cyanobacterial studies than on marine algal studies to reference and discuss this work. This is particularly important considering that the problematic microorganisms are different (red tide such as dinoflagellates, diatoms and other protists in marine environments, versus cyanobacteria, i.e. bacteria, in freshwater), and therefore, perhaps their responses and behaviours also differ.
Concern#4: Connecting to the previous concern, I need to remark that M. aeruginosa is a cyanobacterium (bacteria, prokaryote), not an algae (protists, eukaryote) as the authors pointed out throughout the article.
Concern#5: The author highlighted that Prodigiosin, PQS and the other unknown compounds are produced by the bacterium to kill M. aeruginosa. However, as far I understood from the methodology, the extracts are coming from a pure bacterial culture instead of the cyanobacterial-bacterial co-culture. Therefore, how do the authors demonstrate that the production of the compound occurs in response to the presence of M. aeruginosa cells? Are these algicidal compounds constitutively expressed and produced?
Concern#6: In line 189-190, authors said: “After 7 d of co-cultivation, the algal surface became significantly rugose and the bacteria adhered to the algal cells via their flagella, which showed a mutualism-parasitism relationship shift between algae and bacteria; this can be macroscopically manifested as the inhibition or killing of algae”. How do the authors know that the bacteria adhered to the algal cell via their flagella?
Concern#7: I do not understand why authors relate PQS to laccase activity. This hypothesis should be explained. In addition, the authors noted that PQS enhanced the algicidal activity of E. hormaechei because it is a quorum sensing molecule, but wouldn't it also be possible that PQS acts as an algicide compound in combination with the other substances released by the bacterium?
Concern#8: In the transcriptomic analysis, authors used time 0 as control. However, I wonder if the changes obtained after 4 and 7 days are due to the action of the algicidal bacteria or are related to the culture senescence. I think that a proper control to compare would be the pure cyanobacterial culture, without bacteria, at the same experimental time.
Concern#9: The transcriptomic analysis is key in this study, providing the most interesting results. For this reason, I recommend the authors to explain more deeply (go deeper into the data obtained) and better (explain more clearly, and include further discussion with other related studies) the section 3.3.
Concern#10: In line 435, the authors state that "E. hormaechei F2 detects the presence of M. aeruginosa cells through the QS system and then secretes algicidal compounds", but to draw such a conclusion the authors should adequately explain the results obtained. In the same paragraph (L438-439) it is also said “Simultaneously, E. hormaechei F2 forms biofilms that wrap algal cells” but one thing is to attach to the cell (as figure 1) and different thing is to “wrap” the cells. Please, explain better and use the proper scientific jargon.
Concern#11: The fact that the conclusion ends with "threat to coastal ecosystems" (L456) clearly indicates the wrong focus of the article. If the authors are focusing on a bloom-forming species that threatens inland waters, the focus of the article, including the references cited, should be on freshwater, microcystis genus, and the challenges facing reservoirs and lakes.
On the other hand, minor remarks should be also considered.
- In line 27, authors referenced a study focused on marine environments, while the targeted cyanobacteria is from freshwater. Please include more suited review article.
- In line 31, authors reference an old study again on marine environments and focused on China. I suggest other more recent reviews on the topic that highlight the problem worldwide (for example see Svirčev et al., 2019).
- In line 43, authors wrote that prodigiosin has an algicidal effect on Hahella chejuensis, which is a bacterium not an algae or cyanobacteria, so it could not be called “algicidal” activity.
- In line 132-133, authors said: “Extract 2 and 6 were dissolved in phenazine (Tokyo Chemical Industry, Tokyo) and PQS (Bidepharm, Shanghai) was dissolved in methanol”. Did the authors dissolve the extract 2 and 6 in phenazine? I think there is something wrong because this is the same compound that they found as algicidal.
- In line 162-164, I think the authors forgot some personal clarification.
- In all figures, the figure caption is separate from the main title.
- In line 202, what do the authors mean by “fermentation broth”?
- In figure 3 and 4 there is no explanation about what phenotype images are and represent. Are they an agar plate picture? Or a photo taken from the bottom of the flask?
- In line 238, please provide the reference in “other reported AQs families”.
- In line 243, Where did the number of cells come from and how did the authors calculate it?
- In line 252, what did the authors mean by "PQS could act as a QS molecule that regulates algicidal genes and enzymes"? Does this sentence come from Figure 4?
- There are many empty spaces throughout the article. The authors should take care that the headings of the sections are well placed to facilitate reading in the revision process.
- Why are there section 3.3.1 but not 3.3.2? If there is only one sub-section, please keep only heading 3.3.
References:
Svirčev, Z., Lalić, D., Bojadžija Savić, G., Tokodi, N., Drobac Backović, D., Chen, L., Meriluoto, J., Codd, G.A., 2019. Global geographical and historical overview of cyanotoxin distribution and cyanobacterial poisonings, Archives of Toxicology. Springer Berlin Heidelberg. https://doi.org/10.1007/s00204-019-02524-4
Yang, C., Hou, X., Wu, D., Chang, W., Zhang, Xian, Dai, X., Du, H., Zhang, Xiaohui, Igarashi, Y., Luo, F., 2020. The characteristics and algicidal mechanisms of cyanobactericidal bacteria, a review. World J. Microbiol. Biotechnol. https://doi.org/10.1007/s11274-020-02965-5
Yang, F., Wei, H.Y., Li, X.Q., Li, Y.H., Li, X.B., Yin, L.H., Pu, Y.P., 2013. Isolation and characterization of an algicidal bacterium indigenous to lake taihu with a red pigment able to lyse microcystis aeruginosa. Biomed. Environ. Sci. 26, 148–154. https://doi.org/10.3967/0895-3988.2013.02.009
Zhang, Y., Chen, D., Zhang, N., Li, F., Luo, X., Li, Q., Li, C., Huang, X., 2021. Transcriptional Analysis of Microcystis aeruginosa Co-Cultured with Algicidal Bacteria Brevibacillus laterosporus. Int. J. Environ. Res. Public Health 18, 8615. https://doi.org/10.3390/IJERPH18168615
Reviewer 2 Report
The manuscript is good enough to publish. There are only some minor questions may be needed to clarify. First of all, is the algicides from E. hormaechei F2 species-specific? If not, would it affect the other microorganism within the water body which occur HABs and impact the microbial diversity? Are there any other algicide-related genes rather than the chosen genes? Content in Line 162-164 should be deleted as it comes from the template.
Reviewer 3 Report
The manuscript entitled “Identifying algicides of Enterobacter hormaechei F2 for control of the harmful alga Microcystis aeruginosa” deals with the characterization of algicides (prodigiosin, and PQS-like substances) with different methods including electron microscopy, chromatograph-mass spectrometry and qRT-PCR analysis. Such a combination of methods allowed the authors to obtain results that are of interest to many researchers. I think the manuscript deserves publication on International Journal of Environmental Research and Public Health, however I suggest few revisions:
General concept comments concern some general conclusions of the authors, which may not be fundamental for the article, but show some inaccuracy in the description of the results obtained:
- Line 189 the bacteria adhered to the algal cells via their flagella, which showed a mutualism-parasitism relationship shift between algae and bacteria
Firstly, there are no signs of flagella in the photographs. In addition, I have never come across a mention that bacterial flagella can be used as for locomotion and for attachment. If this is a completely new fact, then it should be supported by literature references or own photographs, where it would be clear that flagella are attached to the surface of microalgae cells. Actually, the presence of flagella in this strain should also be shown, since this is a systematic feature.
Secondly, it is not clear how this fact shows "a mutualism-parasitism relationship shift between algae and bacteria". Such general conclusions should be explained or avoided.
- Line 249 which are characteristic features of programmed cell death [13, 14, 18].
In the works cited, there are no data on programmed cell death. This process is very specific, and although it is described in microalgae, however, these are completely different works. (as Kay D. Bidle, Programmed Cell Death in Unicellular Phytoplankton, Current Biology, Volume 26, Issue 13, 2016, Pages R594-R607, https://doi.org/10.1016/j.cub.2016.05.056. for review, or Marı́a Segovia, Liti Haramaty, John A. Berges, Paul G. Falkowski, Cell Death in the Unicellular Chlorophyte Dunaliella tertiolecta. A Hypothesis on the Evolution of Apoptosis in Higher Plants and Metazoans, Plant Physiology, Volume 132, Issue 1, May 2003, Pages 99–105, https://doi.org/10.1104/pp.102.017129). In order to show the presence of the programmed cell death, more specific methods are required, which do not include SEM and TEM. In the case of this manuscript, with the available data, it is not possible to distinguish programmed cell death from necrosis. A correction should be made - either to remove the mention of programmed cell death, or proof it with specific staining and refer to more relevant works.
Specific comments
Lines 35-36 “The former involves directly attacking algal cells by establishing a connection with the algicidal bacteria flagella”
Direct bacterial attack does not always require a flagellum, here it is necessary to soften the sentence - it is associated with direct contact of the surfaces of bacteria and algae.
Line 147 you should specify which "embedding agent" was used.
Figure 1
It is clear that the highlighting in the photographs is done so that it is clear where the algae are and where the bacteria are. However, since the manuscript is not a popular science article, in my opinion, photos in the classic grayscale form should also be added. Or just indicate algae and bacteria with arrows on the original photos.
Figure 3
It is not clear why Figure 3 has microscopy data for prodigiosin and not for PQS. Are there any explanations for this in the text? If not, they should be added. If data are available but not presented in the article, they should be added, even if they duplicate data for prodigiosin.
Figure S2
Small signatures in the figure are almost unreadable. It is worth using a vector graphics editor to present this drawing and increase the resolution
Reviewer 4 Report
Overall in the present report, Zhang et al. tried to identify the components/compounds which are responsible for the algicidal activity. The study concept seems to be interesting, however, the presentation in the report needs to be majorly revised. Therefore, I would recommend a major revision of the report before publication. The specific points listed below that I found need to be addressed carefully after extensive studies:
General:
English and Typos: Check English and typos carefully throughout the manuscript.
Abbreviations: The full abbreviation of short forms is need to be added at their first appearance; for example: in line 18 there is a word PQS, that should be abbreviated.
Introduction:
Well-written and informative related to the study. However, the description of previously identified compounds or chemical moieties is missing, I recommend incorporating those in 2-3 lines. Also, include the most probable mechanism if reported.
Material and Method:
Section 2.3: The method is totally non-understandable. The authors only described the applications of kits. A detailed method that describes a detailed procedure must be presented. I can only see the name of various Kits here, nothing else. What is the exact study?
Section 2.4: Line 119; Authors wrote; “then centrifuged (5000 g, 10 min, 4℃) to obtain the cell-free bacterial culture.” What is cell-free bacterial culture?? It seems cell-free media (supernatant) right?
In the same section, the authors used the rotavapor to concentrate the media from 1000 mL to 100 mL. Did you use a high temperature of more than 80? If yes, did the author check the stability of the compound? Researchers must try lyophilization to avoid any temperature-based destruction of compounds.
In the same section, researchers extracted the cell culture supernatant with pet ether, chloroform, ethyl acetate, and methanol and reported the highest activity in methanolic extract. It is not possible to do methanolic extraction of water as both are miscible. These experiments and related results seem highly ambiguous. The same is with the pet ether. Clarify this
In the same section, What is 15 drops/min? this is unscientific. The flow rate must be in ml/min or ul/min.
What is extract 2 and extract 6? Redo this experiment and write properly.
Section 2.5: Define extracts 2 and 6 before.
The researchers are doing LCMS directly before knowing the compound’s property. I don’t understand how an LCMS study could be done before knowing the exact mass of the compound. For identifying, the compound structure and other properties NMR is the most acceptable technique. Then the functional groups are determined by using the FTIR and UV vis spectroscopy study. Then we generally move to the Mass analysis. At last, we perform the LCMS study for qualitative and quantitative analysis of the known compound. I don’t understand how authors are identifying unknown compounds using LCMS only? I suggest doing a complete study.
Section 2.7 SEM: The Algal cells are only treated with the bacterial culture and were imaged by SEM. However, there is no experimentation with the extracted compounds. I suggest doing the relevant study with extracted compounds.
Apart from these: there are no positive controls have been studied. The study should be supported with at least one positive control (available).
Results and Discussion:
Methods represent the overall view of the study of what has been done, thus I would recommend incorporating the corrected and detailed study and furnishing the results and the relevant discussion into the manuscript. However, I will comment on some points of the results and the discussion.
Line 162-164, these are general Journal guidelines that must be deleted.
Flow is weird: I don’t understand the flow of results. In methods the flow was different and here it started with SEM analysis after the algicidal effect. I think the focus must be on the identification of the compounds first.
Figure 1: A: general practice to plug the culture flask is a cotton plug, how can you say you have avoided the contamination here?
The unknown algicidal compounds must be reported and characterized using, NMR, IR, MS, UV-Vis, etc.
Figure 2. the quality is not good, I can not see B and C figures even zooming at 200%.
What algicidal genes or enzymes are exactly affected?
Figure 6. Gene function analysis graphs are too dense, try to summarize and present again with smaller data.
Figure 7. What exactly information we can expect from the transcript profiling and how this study is related to the current manuscript?
Conclusion
It is difficult to conclude the exact identification of algicidal compounds in this study.
Reviewer 5 Report
Please split the results from the discussion.
Please make a section in the methodology about the statistical analysis.
All tables MUST be self-explanatory. Please correct.
All Figures MUST be self-explanatory. Please correct.
This paper must be read and corrected for English editing by native speaking person.
Arrange key words in alphabetical order.
I must revise the paper again after doing English editing.
This must be done: Please split the results from the discussion.
Round 2
Reviewer 1 Report
Response to Authors – Round 2
I thank the authors for considering some of the recommendations made in the first round of review. However, I think there are still important issues to resolve before accepting the article for publication in the journal. Please consider the following concerns:
Concern #1:
- Round 1: One of the most important weaknesses of the manuscript is the poorly elaborated and explained methodology. Many crucial data are missing, making it difficult to understand how and why the experiments are performed. The most striking case is that they provide almost no information on gene expression and transcriptomic analysis. For example, compare with the following paper on the topic recently published in the same journal: Zhang et al., 2021. Other important missing information is how the authors performed the laccase activity assays or how they calculated the number of cells (Figure 4). Nothing from line 256-168 about adding PQS to the co-culture is explained in the material and method. New hypotheses appear throughout the article, but these should be highlighted and based on the main objectives at the end of the introduction.
Response: As requested by the reviewer, more detailed information of gene expression and transcriptomic analysis has been added, including methods and materials (section 2.3) and section 3.3.1, Figure S1.
- Round 2: The material and methods section is still incomplete. For example, there is no information on the experimental flow where the authors decide to add various concentrations of PQS (only a brief explanation appears in L284-287). I understand that this is an idea that arose when they observed that strain F2 produced PQS but this did not affect the growth of Microcystis, but there is relevant information that should be provided, such as: what PQS do they add? Commercial? Pure extract? In my opinion this information should be provided in material and methods as a new section or included in one of the existing sections such as 2.2 Algicidal Activity.
Concern#2:
- Round 1: The study is missing many important references, including the previous study in which the authors isolated Enterobacter hormaechei strain F2. There are many other references that are key to explaining the results but do not appear. For example, the earlier study where a Prodigiosin-producing bacterium was found to lyse M. aeruginosa cells (Yang et al., 2013), or the above commented study focused on transcriptomic analyses (Zhang et al. 2021). Also, the authors provided references focused on marine environments (i.e. L48-55), while the target cyanobacterium is mainly problematic and common in freshwater. Microcystis is the most studied bloom-forming genus of cyanobacteria, and there are more than 50 studies demonstrating the algicidal activity of various bacteria and compounds on it (for instance, see the last review on the topic: Yang et al., 2020). In my opinion, the authors should focus more on freshwater cyanobacterial studies than on marine algal studies to reference and discuss this work. This is particularly important considering that the problematic microorganisms are different (red tide such as dinoflagellates, diatoms and other protists in marine environments, versus cyanobacteria, i.e. bacteria, in freshwater), and therefore, perhaps their responses and behaviours also differ.
Response: We thank the reviewer for the careful examination of our reference list. As requested by the reviewer, we modified the references to make them more in line with the theme of our article.
- Round 2: The authors have not taken into account any of my recommendations to this comment. They have added neither the reference of the previous article of F2, nor that of the other Prodigiosin producing bacteria against M. aeruginosa, nor the previous study on transcriptomic analysis. They have also added three new references, again focused on marine microorganisms. In fact, as I commented in round 1, I still think that the article is very focused on microoganisms that affect the marine environment, while the results provided focus on M. aeruginosa, a cyanobacterium that is problematic mainly in freshwater.
Concern#3:
- Round 1: The author highlighted that Prodigiosin, PQS and the other unknown compounds are produced by the bacterium to kill M. aeruginosa. However, as far I understood from the methodology, the extracts are coming from a pure bacterial culture instead of the cyanobacterial-bacterial co-culture. Therefore, how do the authors demonstrate that the production of the compound occurs in response to the presence of M. aeruginosa cells? Are these algicidal compounds constitutively expressed and produced?
Response: The reviewer raised an important point and we thank the reviewer for this comment. Indeed, as mentioned by the reviewer, these algicidal compounds are constitutively expressed and produced. Data from another study of us showed that the concentration of these algicidal compounds increased significantly after the co-cultivation of bacteria and algae, suggesting that the production of the algicidal compounds were stimulated by algae at the transcriptome and metabolome level.
- Round 2: Thank you for this clarification. I think that this information is key, so it must be highlighted in manuscript and mentioned reference should be included.
Concern#4:
-Round 1: I do not understand why authors relate PQS to laccase activity. This hypothesis should be explained. In addition, the authors noted that PQS enhanced the algicidal activity of E. hormaechei because it is a quorum sensing molecule, but wouldn't it also be possible that PQS acts as an algicide compound in combination with the other substances released by the bacterium?
Response: It is reasonable to infer that PQS functions as quorum sensing molecule based on references and experimental phenomena. With the presence of PQS at a concentration of 2 μM, the activity of laccase became significantly higher. Besides, the expression of three algicidal related genes K00059, K00652, and K01657, where up-regulated, indicated PQS participates in the algicidal activity as a QS molecule that regulates algicide-related enzyme activity and gene expression.
- Round 2: I understand that the authors study PQS as a quorum sensing molecule because it has been widely described as such, but what is not clear is how they came up with the idea of measuring laccase activity and relating it to PQS. Is it because it is usually measured in other microorganisms? Is it because it is usually measured in other microorganisms? I noticed that authors included a reference to its production in fungi, did the idea come from this study? I am not saying that it is wrong to measure laccase activity, what I am saying is that the reasoning behind the authors' decision to measure it is lacking. As the current version stands, it seems to have occurred to them suddenly and fortunately they have found a link. This result may be important, so I recommend the authors to elaborate more on the explanation. For example, it is not even known how they measured laccase activity because it is not explained in material and methods.
Concern#5:
- Round 1: 10. In line 243, Where did the number of cells come from and how did the authors calculate it?
Response: The method is added to section 2.2, as “M. aeruginosa cells were counted by hemocytometer plate, and the average value of three parallel samples was taken as the cell number .”
- Round 2: Thank you for this clarification. However, I think the authors should include the formula they used to calculate the algicidal ratio or at least add a proper reference.
Concern#6:
- Round 1: In figure 3 and 4 there is no explanation about what phenotype images are and represent. Are they an agar plate picture? Or a photo taken from the bottom of the flask?
Response: The phenotype images in this research were taken from the bottom of the flask.
- Round 2: Thank you for this clarification. Please add this information in material and method or figure caption.
Concern#7:
The reference numbers in the text do not match the number provided in the reference section. Please check and modify accordingly. Also, in L118, the added sentence is not well explained (e.g., see "toke").
Comment:
- Round 1: In the transcriptomic analysis, authors used time 0 as control. However, I wonder if the changes obtained after 4 and 7 days are due to the action of the algicidal bacteria or are related to the culture senescence. I think that a proper control to compare would be the pure cyanobacterial culture, without bacteria, at the same experimental time.
Response: Many thanks to the reviewers for the suggestions, which provided us with valuable ideas for the subsequent experiments. In fact, the focus of our study was on the transcriptome of the bacterium, such that cyanobacteria at 0 moments were not suitable as a control for bacterial transcriptome data. But starting from the transcriptome of cyanobacterial is not lacking as a novel starting point, thank you again for the ideas you provided
- Round 2: I am sorry for this confusion. It was my mistake and the question did not make sense. My apologies.
Reviewer 4 Report
The authors did only superficial changes and ignored the suggestions in most of the answers. The most important thing that the author should care about is the science that is already established and known to everyone. Those mistakes must be avoided, such as, in the previous round of questions, I asked how contamination of the flasks was avoided without using cotton plugs? They answered, “parafilm was used to cover the flask during autoclaving”. Nobody can trust this, how is parafilm can sustain during autoclaving? It’s not possible at all while the melting point of parafilm is max 66 °C. If the flow and concept are for transcriptomics why title starts from “Identifying algicides” it requires additional purification, and identification confirmation using NMR and FTIR techniques. I would suggest changing it to “Transcriptome based identification of algicides from ………..”. Overall, after significant improvement only It could be reconsidered.
Author Response
Please see the attachmen

Reviewer 5 Report
Please make a section in the methodology about the statistical analysis.
This should be a separate title under the materials an methods. it must be separate section,. I must revise the paper again.
Round 3
Reviewer 5 Report
The paper can NOW be accepted in its present format .